# Subgrid-scale variability of cloud ice in the ICON-AES 1.3.00

Sabine Doktorowski[1], Jan Kretzschmar[1], Johannes Quaas[1], Marc Salzmann[1], and Odran Sourdeval[2]

[1]Institute for Meteorology, Universität Leipzig, Leipzig, Germany
[2]Laboratoire d'Optique Atmosphérique, Université de Lille, Lille, France

**Correspondence:** Sabine Doktorowski (sabine.hoernig@uni-leipzig.de)

**Abstract.** This paper presents a stochastic approach for the aggregation process rate in the ICON-AES, which takes subgrid-scale variability into account. This method creates a stochastic parameterisation of the process rate by choosing a new specific cloud ice mass at random from a uniform distribution function. This distribution, which is consistent with the model's cloud cover scheme, is evaluated in terms of cloud ice mass variance with a combined satellite retrieval product (DARDAR) from the satellite cloud radar CloudSat and cloud lidar CALIPSO. The global patterns of simulated and observed cloud ice mixing ratio variance are in a good agreement, despite an underestimation in the tropical regions, especially at lower altitudes, and an overestimation in higher latitudes from the modeled variance. Due to this stochastic approach the yearly mean of cloud ice shows an overall decrease. As a result of the non-linear nature of the aggregation process, the yearly mean of the process rates increases when taking subgrid-scale variability into account. An increased process rate leads to a stronger transformation of cloud ice into snow and therefore, to a cloud ice loss. The yearly averaged global mean aggregation rate is more than 20 % higher at selected pressure levels due to the stochastic approach. A strong interaction of aggregation and accretion, however, lowers the effect of cloud ice loss due to a higher aggregation rate. The presented new stochastic method lowers the bias of the aggregation rate.

## 1 Introduction

A correct representation of clouds and cloud related microphyscial process rates, which describes the time dependent source and sink terms of cloud ice or liquid water, is one of the central challenges in global climate modelling. Since global climate models typically run on a rather coarse resolution (order of 100 km), it is important to look into the unresolved microphysical process rates. Most climate models use a cloud cover parameterization. Microphysical process rates are computed based on grid box mean in-cloud ice/liquid water mixing ratios. In-cloud cloud ice mass mixing ratio is the cloud ice mass per cloudy area. Considering subgrid-scale variability of in-cloud variables reduces the biases of the non-linear microphysical process rates (Pincus and Klein, 2000; Larson et al., 2001). For example, Weber and Quaas (2012) numerically integrated the process rate over the probability density function (PDF), which is a very accurate method but needs additional computational time. Another method, which works with no additional computational time, is a stochastic approach for the process rates by taking a randomly chosen value per time step and grid-box. (e.g., Palmer, 2001; Berner et al., 2017). With this method a randomly disturbed process rate is created in order to give a better representation of the state of atmosphere. Another method is the Cloud Layers Unified By Binormals (CLUBB) (Golaz et al., 2002a, b), which works with a set of PDFs for all cloud types to

avoid difficulties in coupling of stratocumulus and shallow convection parameterizations. Since CLUBB does not parameterize subgrid-scale variability of cloud ice, Thayer-Calder et al. (2015) includes cloud ice to the CLUBB PDF's. The PDFs are sampled by the Subgrid Importance Latin Hypercube Sampler (SILHS), which connects CLUBB with the microphysics for
stratiform and convective clouds. They found improvements e.g. of liquid water path (LWP), precipitable water and shortwave cloud forcing, but also a degradation in precipitation. Including the subgrid-scale effect in the autoconversion and accretion rate in warm clouds reduces the bias significantly and leads to an enhancement of the process rate (Boutle et al., 2014; Lebsock et al., 2013). Since previous studies mainly focus on warm rain formation processes (e.g., Morrison and Gettelman, 2008; Larson and Griffin, 2013; Boutle et al., 2014; Lebsock et al., 2013), it is also important to concentrate on snow formation
effects, by taking subgrid-scale effects into account.

Mülmenstädt et al. (2015) emphasized that most of the global rain is produced via ice phase processes. Thus, we focus on cloud ice related process rates, especially on the precipitation processes initiated via the ice phase. In the ICON-AES snow is formed by the aggregation process. Aggregation describes the process where cloud ice particles grow to snowflake sizes by sticking together. Therefore, in this study we implement a stochastic aggregation parameterization into the Icosahedral Nonhydrostatic
general circulation model (ICON-AES, Giorgetta et al., 2018) by taking subgrid-scale variability of cloud ice into account. Subgrid-scale variability of the total water mixing ratio (sum of cloud liquid water, cloud ice and water vapour) is already used for determining the cloud cover (Sundqvist et al., 1989). Here, we use this uniform distribution approach to create a distribution of cloud ice within the cloudy part of the grid box. Instead of taking a grid-box mean in-cloud ice mixing ratio for the non-linear aggregation parameterization, we feed the process rate with a cloud ice mass randomly chosen from the distribution of cloud
ice mass assumed in the cloud scheme.

To evaluate the uniform distribution of cloud ice at a global scale, large-scale observations of cloud ice are necessary. The combined data set of spaceborne radar from the CloudSat satellite (Stephens et al., 2002) and the lidar from the Cloud-Aerosol Lidar and Infrared Pathfinder Observations satellite (CALIPSO, Winker et al., 2010) allows to retrieve a global data set of cloud ice. Here, we use the liDAR - raDAR (DARDAR Delanoë and Hogan, 2008, 2010) dataset. Comparing observed cloud
ice with modeled cloud ice is a challenge. Since the cloud ice from the ICON-AES does not include snow (precipitating cloud ice) and convective cloud ice, it is necessary to remove falling and convective cloud ice from the DARDAR data set in order to give a meaningful comparison between model and observations. A flag method from Li et al. (2008, 2012) is used to estimate the precipitating and convective cloud ice part per grid-box, which is removed from the data set.

In this study, we investigate an important process rate which transforms ice to snow, the aggregation rate, and how it is treated
in the ICON-AES. We include the stochastic approach into the aggregation in order to quantify the influence of taking subgrid-scale variability into account. The selected distribution function of cloud ice is evaluated with the DARDAR data set and the effect of the stochastic approach in the ICON-AES is investigated. As an additional evaluation, we compare an unbiased process rate with the stochastic approach in order to investigate, how well this simple method performs.

## 2   Methods

For all simulations the ICOsahedral Nonhydrostatic general circulation model (ICON-AES-1.3.00, Giorgetta et al., 2018) in its global version is used. It includes the Max Planck Institute physics package based on the ECHAM6 physics (Stevens et al., 2013). All runs were performed for six years with prescribed sea surface temperature and sea ice boundary conditions for a period from 2004 to 2009 and with instantaneous diagnostics output every six hours by using a model time step of 10 minutes, a horizontal resolution of 160 km and 47 vertical hybrid sigma levels up to 80 km height (Crueger et al., 2018; Giorgetta et al.,

2018). To avoid any effect of the model spin-up, we ignored the first year from the model results. Therefore all multiyear averages were done for the time period 2005-2009. Afterwards, the ICON-AES data are interpolated to selected pressure levels. The ICON-AES contains prognostic equations for water vapor and for cloud liquid water and cloud ice in stratiform clouds. Stratiform cloud cover is computed by using a diagnostic cloud cover scheme (Sundqvist et al., 1989). Stratiform cloud microphysics is parameterized following Lohmann and Roeckner (1996). The prognostic equation for grid-box mean cloud ice

mixing ratio ($\overline{q}_i$) including the different process rate terms ($Q$) is written as follows

$$\frac{\mathrm{d}\overline{q}_i}{\mathrm{d}t} = Q_{Ti} + Q_{sed} + Q_{dep} - Q_{mli} - Q_{sbi} + Q_{fr} - Q_{saci} - Q_{agg} \tag{1}$$

including advection, parameterized turbulent diffusion, and convective detrainment of cloud ice $Q_{Ti}$, sedimentation of cloud ice $Q_{sed}$, deposition and sublimation $Q_{dep}$, melting of cloud ice $Q_{mli}$, instantaneous sublimation in the cloud free part $Q_{sbi}$, homogeneous and heterogeneous freezing of liquid water $Q_{fr}$, accretion of cloud ice by snow $Q_{saci}$ and the aggregation of

cloud ice $Q_{agg}$ (Giorgetta et al., 2013). Microphysical processes are determined from in-cloud cloud ice and liquid water, respectively. The in-cloud values are calculated by dividing the grid-box mean cloud ice mixing ratio and cloud liquid water mixing ratio by the fractional cloud cover (C). C is calculated by a diagnostic cloud cover scheme by Sundqvist et al. (1989) (more details in section 2.2). Cloud liquid water and cloud ice are considered as well mixed, if they coexist. Therefore, there are no separate liquid or ice parts in the cloud. The ECHAM6 physics includes diagnostic rain and snow profiles in the columns.

It is not transported by advection (Giorgetta et al., 2018).

### 2.1   Aggregation parameterization in the ICON-AES

In this study, we focus on the aggregation parameterization. In the current version of the ICON-AES, with ECHAM6 physics, the conversion rate of ice to snow by aggregation as a non-linear process is given by Levkov et al. (1992), based on the work of Murakami (1990), with

$$Q_{agg} = \frac{q_i}{\Delta t_1}, \tag{2}$$

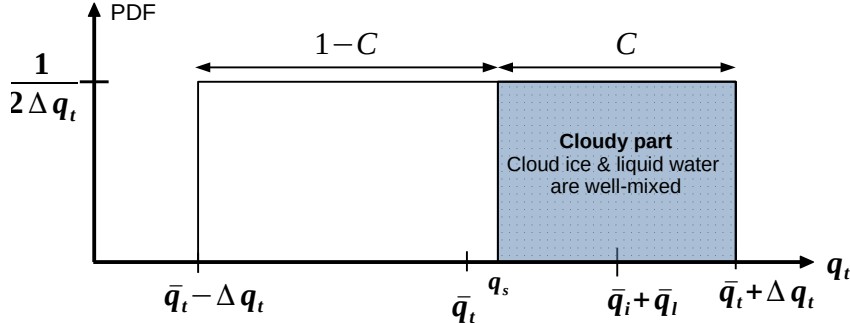

**Figure 1.** Schematic overview of cloud ice variability as part of the cloud cover parametrization: The cloud cover is defined as the part of PDF which exceeds the saturation humidity $q_s$ up to $\bar{q}_t + \Delta q_t$. Within the cloudy part we define a distribution over the hydrometeors $q_i + q_l$. The half width of the PDF of $\bar{q}_i$ is defined as the C multiplied with $\Delta q_t$ and the cloud ice fraction ($f_{ice} = \frac{\bar{q}_i}{\bar{q}_i + \bar{q}_l}$)

$\Delta t_1$ is defined as the time, which is needed for ice crystals to grow from the mean radius $\overline{R}_{\mathrm{vi}}$ to the smallest radius of snow class particles $R_{S0}$.

$$\Delta t_1 = -\frac{2}{c_1} \log\left(\frac{R_{\mathrm{vi}}}{R_{S0}}\right)^3,$$ (3)

with

$$c_1 = \frac{q_i \rho a_{\mathrm{I}} E_{\mathrm{ii}} X}{\rho_i} \left(\frac{\rho_0}{\rho}\right)^{\frac{1}{3}},$$ (4)

where $a_{\mathrm{I}} = 700\,\mathrm{s}^{-1}$ is an empirical constant, $E_{\mathrm{ii}} = 0.1$ the collection efficiency, $\rho_i = 500\,\mathrm{kg\,m^{-3}}$ the density of cloud ice and $\rho_0 = 1.3\,\mathrm{kg\,m^{-3}}$ the reference density of air. Combing equations (2) - (4) leads to the final aggregation rate equation:

$$Q_{\mathrm{agg}} = C\gamma \frac{\rho \bar{q}_i^2 a_{\mathrm{I}} E_{\mathrm{ii}} X \left(\frac{\rho_0}{\rho}\right)^{\frac{1}{3}}}{-2\rho_i \log\left(\frac{R_{\mathrm{vi}}}{R_{\mathrm{s0}}}\right)^3}.$$ (5)

The process may be tuned with a tuning parameter $\gamma$, which is currently set to $\gamma = 95$. The parameterization uses the grid-box mean in-cloud ice mixing ratio to calculate the process rate of aggregation from ice to snow, which introduces biases in the process rates, since the aggregation is a non-linear process (Pincus and Klein, 2000). An unbiased aggregation rate is calculated by using an integral over a distribution function of subgrid-scale cloud ice mixing ratio ($\overline{Q_{\mathrm{agg}}(q_i)}$).

## 2.2 Subgrid-scale variability of cloud ice in the aggregation process

The ICON-AES determines the fractional cloud cover according to Sundqvist et al. (1989). A uniform distribution function of the total water mixing ratio $q_t$ from $\bar{q}_t - \Delta q_t$ to $\bar{q}_t + \Delta q_t$ is considered (Figure 1). The total water mixing ratio describes the

sum of water vapour, cloud liquid water and cloud ice mixing ratio. The saturation specific humidity $q_s$ is calculated from the grid-box mean temperature considering the saturation with respect to ice at temperatures below 0°C and if $q_i$ is higher than a threshold value $\gamma_{\text{thr}} = 5 \cdot 10^{-7}$ kg/kg. The integral over the distribution from the saturation humidity $q_s$ up to the maximum of the distribution function $\overline{q}_t + \Delta q_t$ defines the fractional cloud cover C, which depends on the calculation of the distribution width $\Delta q_t = \gamma q_s$. The parameter $\gamma$ varies with height from low values near the surface to larger ones in the free troposphere with a prescribed profile (Quaas, 2012). The final equation for C, which is used in the model, is given by

$$C = 1 - \sqrt{1 - \frac{r - r_0}{r_{\text{sat}} - r_0}}, \tag{6}$$

where $r$ is the relative humidity, $r_{\text{sat}}$ is the saturation value (= 1) and $r_0$ is a function of pressure and depends on two different tuning parameters ($r_{\text{top}} = 0.8$ and $r_{\text{surf}} = 0.968$), which defines the condensation threshold.

To define a new subgrid-scale cloud ice mass, we use this fractional cloud cover approach (Figure 1). The half width of the cloud ice PDF ($\Delta q_i$) has to be re-scaled with $\Delta q_i = C \Delta q_t f_{\text{ice}}$, where $f_{\text{ice}}$ describes the cloud ice fraction ($\frac{\overline{q}_i}{\overline{q}_i + \overline{q}_l}$). Using the PDF over $q_t$ the all-sky cloud ice and cloud liquid water mixing ratio is defined as the integral over $q_t - q_s$ from $q_s$ to the maximum of the PDF($q_t$):

$$\overline{q}_i + \overline{q}_l = \int_{q_s}^{\overline{q}_t + \Delta q_t} (q_t - q_s) \text{PDF}(q_t) \text{d}q_t. \tag{7}$$

Here, $q_i$ and $q_l$ are considered as well-mixed within the cloudy part of the grid box, so $q_i$ and $q_l$ occur in the same volume. Solving the equation (7) yields:

$$\overline{q}_i + \overline{q}_l = C^2 \Delta q_t. \tag{8}$$

To get $\overline{q}_i$, equation (8) is multiplied with $f_{\text{ice}}$, which leads to:

$$\overline{q}_i = (\overline{q}_i + \overline{q}_l) \cdot f_{\text{ice}} = C^2 \Delta q_t \cdot f_{\text{ice}}. \tag{9}$$

As described above, the in-cloud cloud ice mixing ratio $q_i^c$ is defined as $q_i$ divided by C and in combination with equation (8) it follows, that:

$$\overline{q}_i^c = \frac{\overline{q}_i}{C} = C \Delta q_t \cdot f_{\text{ice}} = \Delta q_i. \tag{10}$$

This implies, that the width of cloud ice distribution corresponds to the in-cloud grid-box mean cloud ice mixing ratio $\overline{q}_i^c$. To obtain the subgrid-scale cloud ice mixing ratio, we use a Monte Carlo approach. Here, we choose an element from the cumulative distribution function (CDF) with the help of a random number $r \in [0, 1]$. This yields the following equation:

$$q_{i,\text{new}}^c = \overline{q}_i^c + \Delta q_i (2r - 1). \tag{11}$$

with (10)

$$q_{i,\text{new}}^c = \overline{q}_i^c (2r). \tag{12}$$

Finally, the subgrid-scale cloud ice mixing ratio only depends on the grid-box mean cloud ice mass mixing ratio and the choice of the random number. This new specific cloud ice mass replaces the grid-box mean cloud ice in Equation 5. To compare the current, biased aggregation rate $(Q_{\text{agg}}(\overline{q}_i))$ with the unbiased process rate $(\overline{Q_{\text{agg}}(q_i)})$ we replace for each time step and grid box $\overline{q}_i$ mean in equation (5) with the integral over the entire distribution of $q_i$. This comparison will be shown in the last part of the results.

To evaluate the distribution function, the cloud ice variance is calculated for all-sky conditions, because all output cloud variables are also all-sky variables. The variance can be written as follows:

$$\sigma_{q_i}^2 = \int\limits_0^{2\Delta q_i} (q_i - \overline{q}_i)^2 \, \text{PDF}(q_i) \mathrm{d}q_i. \tag{13}$$

As we focus on all-sky variance, this equation can be separated into two parts: the clear sky part of the distribution $(q_i = 0)$ and the cloudy part $(q_i > 0)$. The cloud free part can be described with Dirac's delta function, which yields the equation (derivation is shown in the appendix):

$$\sigma_{q_i}^2 = \int\limits_0^{2\Delta q_i} \left[ (1 - \mathrm{C}) \, \delta(q_i = 0) + \mathrm{C} \, \frac{1}{2\Delta q_i} \right] (q_i - \overline{q}_i)^2 \mathrm{d}q_i$$

$$\sigma_{q_i}^2 = (q_i^c)^2 \cdot \left( \frac{4}{3} \mathrm{C} - \mathrm{C}^2 \right). \tag{14}$$

## 2.3 Satellite retrievals of cloud ice water content

As mentioned above, we will focus on ice phase processes and their impact on the global ice water content. To evaluate the results, a combined global ice water product of the CloudSat (Stephens et al., 2002) and Cloud-Aerosol Lidar and Infrared Pathfinder Satellite Observations (CALIPSO) (Winker et al., 2010), the DARDAR-Nice data set (Sourdeval et al., 2018), is used. Both satellites are part of the A-Train constellation and are flying with a time interval of only 15 s between them. The W-Band (94 GHz) cloud profiling radar (CPR) on CloudSat provides vertical cloud profiles with a minimum detectable reflectivity of -28 dBz. CALIPSO contains the Cloud-Aerosol Lidar with Orthogonal Polarization (CALIOP) which measures coaligned pulses of 532 nm and 1064 nm wavelength. Both satellite instruments provide global retrievals of clouds and their properties. A combined data set of lidar and radar measurements provides global information about larger as well as smaller ice particles and therefore a more detailed overview of the cloud. Outside the temperature range of -40°C to 0°C, supercooled liquid water does not exist. The used time period of the data set is from January 2007 until December 2010, since the data sets are available for the entire years.

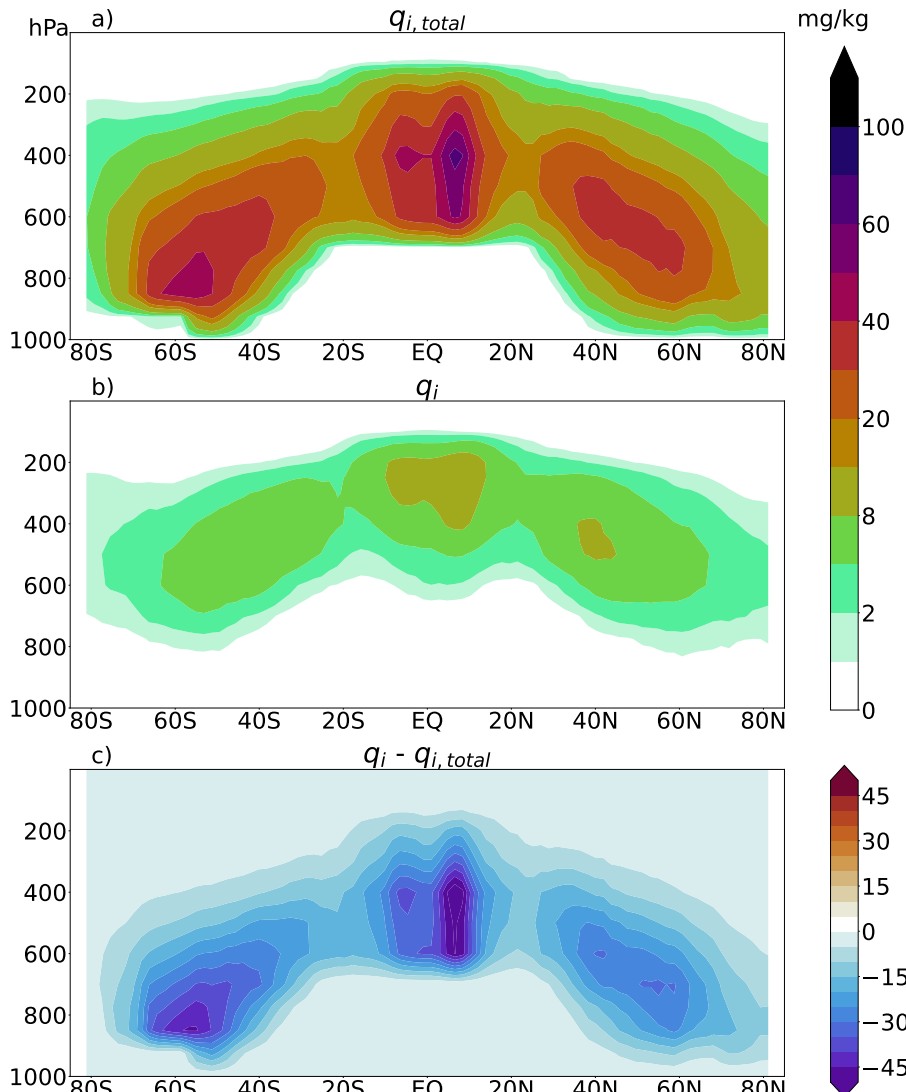

**Figure 2.** Zonally averaged annual mean of ice water content (mg/kg) from the DARDAR data set. a) total ice mixing ratio ($q_{i,total}$), which includes cloud ice from any clouds, and precipitating ice; b) cloud ice mixing ratio ($q_i$), where precipitating and convective cloud ice are removed; c) difference between $q_{i,total}$ and $q_i$.

For an accurate comparison between satellite data and model data the two-dimensional DARDAR data has to be interpolated to the same horizontal grid as the ICON-AES. For each grid-box, the mean of all variables is calculated and vertically interpolated to the same pressure levels as for the ICON-AES. The modeled $q_i$, which is used for the aggregation parameterization, does not include precipitating and convective cloud ice. Therefore, an estimate of the convective and precipitating cloud ice has to be removed from the satellite data set. To provide a data set that is comparable with the ICON-AES, a flag-method after Li

et al. (2008, 2012) is used. All columns that are flagged as "precipitating at the surface" are not considered, to ensure that larger falling ice particles are removed from the data set. This is because the model distinguishes between precipitating ice and cloud ice in two bulk classes. In contrast, it is challenging for the satellite retrievals to separate falling ice and cloud ice. Additionally, cloud ice, which is flagged as originating from "deep convection" or "cumulus" (2B-CLDCLASS data set), is also removed. In order to compare modeled cloud ice with observations, alternative methods are possible (e.g. removing ICON non-zero surface precipitation points,...). Since we want to evaluate the cloud ice distribution, that is used for the aggregation, we had to adjust the DARDAR data in order to find the most consistent way.

Cloud variables simulated by the ICON-AES are output for grid-box-mean, all-sky conditions. Therefore, grid-box mean cloud ice and cloud ice variance are calculated for all-sky conditions in the observational data as well.

Figure 2 shows a multiyear mean of zonally averaged total ice mixing ratio ($q_{\mathrm{i,total}}$) and cloud ice mixing ratio ($q_{\mathrm{i}}$) obtained from the DARDAR product. $q_{\mathrm{i,total}}$ shows the highest values over the equator at lower pressure fields and over the mid-latitudes at higher pressure fields. Most of the removed cloud ice is detected due to the precipitation flag. As a result, the remaining cloud ice water content (Fig. 2b) is much lower, especially in the the mid-latitudes at lower regions and in the tropics. The maxima of cloud ice are shifted upward, since most of the precipitating cloud ice is below or in the lower regions of the cloud. The plot highlights the importance of removing precipitation from the DARDAR data set in order to give a more reliable comparison with the ICON-AES. This is due to the fact, that the remaining $q_{\mathrm{i}}$ is much less than $q_{\mathrm{i,total}}$. But we have to keep in mind, that the result of $q_{\mathrm{i}}$ strongly depends on the accuracy of the different flags, because it selects which cloud ice is detected as convective or precipitating cloud ice and therefore, which cloud ice is removed from the dataset. Additionally, the initial satellite data are measured on a 1D curtain, while the model uses 3D grid boxes. Hill et al. (2015) calculated a measure of the difference in standard deviation considering a 2D cloud field compared to a 1D cross-section. They estimated a 30% larger standard deviation in 2D fields compared to the 1D track. Therefore, we should also consider, that this has an effect also on our the cloud ice variance calculation. However, there are limitations in availability of satellite data. Therefore we tried to be consistent as possible in the comparison between simulations and observations.

## 3 Results

### 3.1 Employing a distribution of cloud ice in the aggregation parameterization of the ICON-AES

As described above, for the evaluation of the cloud ice distribution the all-sky cloud ice variance is calculated for each grid-box. Equation (14) was used for the modeled cloud ice variance, while for the DARDAR data the spatial variance within each GCM grid box was calculated from all footprints within a grid-box. But before comparing the cloud ice variance, we have to make sure, that the cloud ice mixing ratio from DARDAR data and ICON-AES show the same order of magnitude. Figure 3 shows annually averaged global distribution of cloud ice from DARDAR and ICON and the differences of them at different pressure levels. ICON-AES shows higher values in the midlatitudes at higher pressure fields. At 400 hPa there is the main difference around the equator, where DARDAR shows higher values than ICON-AES, ICON-AES still has the maximum values in the

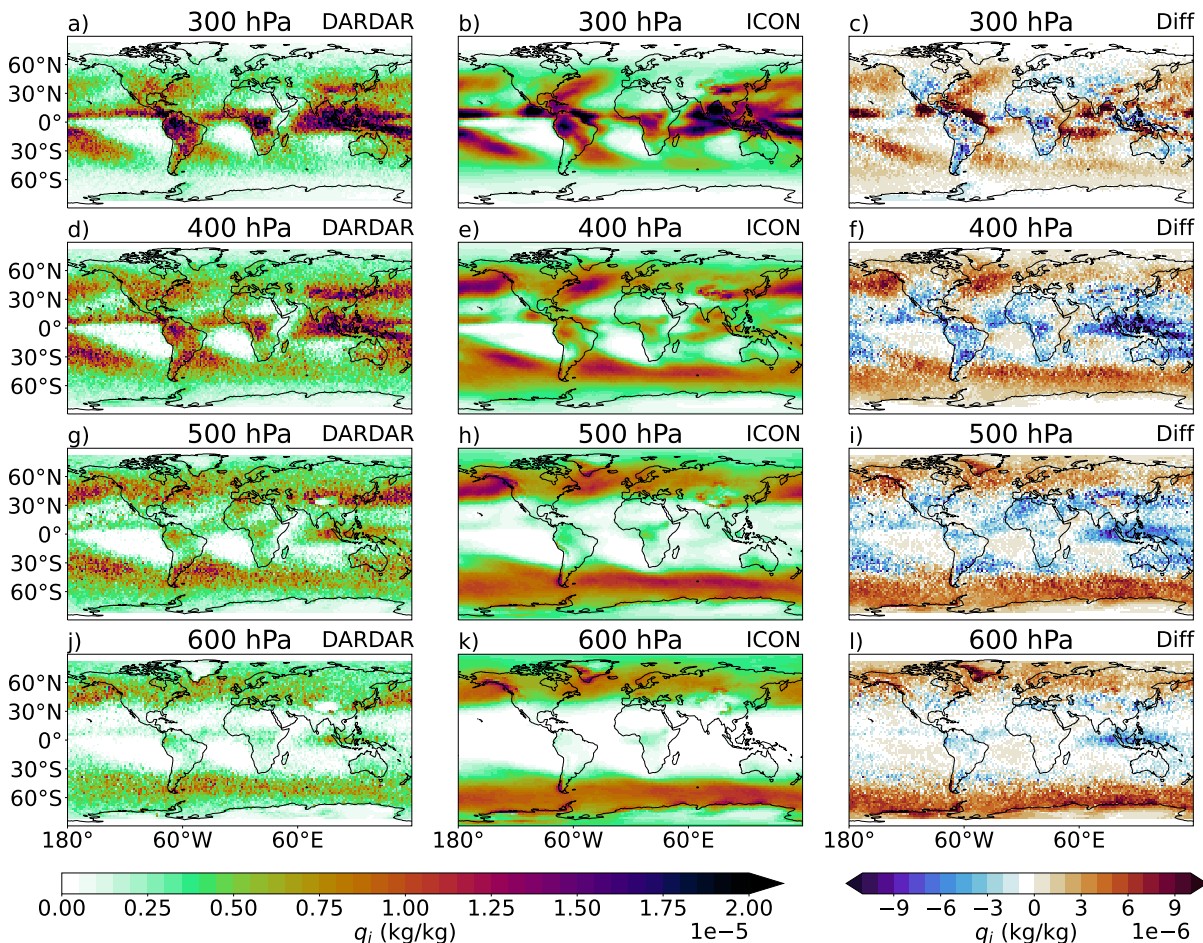

**Figure 3.** Multiyear mean of the cloud ice mixing ratio (kg kg$^{-1}$) at four different pressure levels calculated for the DARDAR data (a,d,g,j), the ICON data (b,e,h,k) and diference between ICON and DARDAR (c,f,i,j).

midlatitudes. However, there is a good agreement between in the pattern of cloud ice mixing ratio of ICON and DARDAR.

The cloud ice water path (CIWP), which is the column integrated ice water content, is given in figure 4. ICON overestimates the CIWP in the middle and higher latidues, while it underestimates the CIWP in the tropics. As it was already visible in the figure 3 DARDAR shows larger cloud ice values down the lower altitudes over the tropics, which leads to an increased CIWP

compared to the modeled CIWP. Especially in the midlatidues at higher pressure fields the model tend to overestimate the cloud ice. One should keep in mind, that the way how the DARDAR data is filtered to get the CIWP or $q_i$, which is comparable with the model data, is not perfect.

Figure 5 shows the comparison of cloud ice variance between ICON-AES and DARDAR data for the same pressure levels chosen in Figure 3. In general, the pattern of cloud ice variance follows the pattern of the global cloud ice distribution. Higher

values are visible over the tropics at higher altitude, and over the storm tracks in the mid-latitudes at lower altitudes. Besides

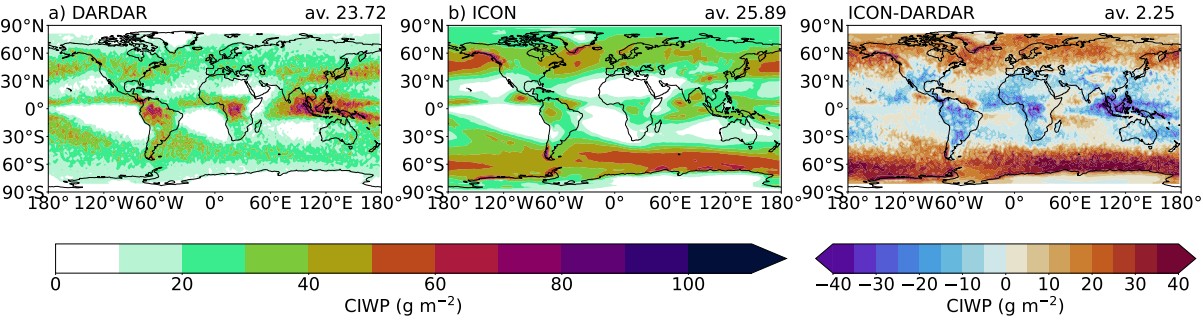

**Figure 4.** Multiyear cloud ice water path (CIWP) for the DARDAR data and ICON data and the difference between DARDAR and ICON

**Figure 5.** Same as figure 3 but for cloud ice variance (kg$^2$ kg$^{-2}$)

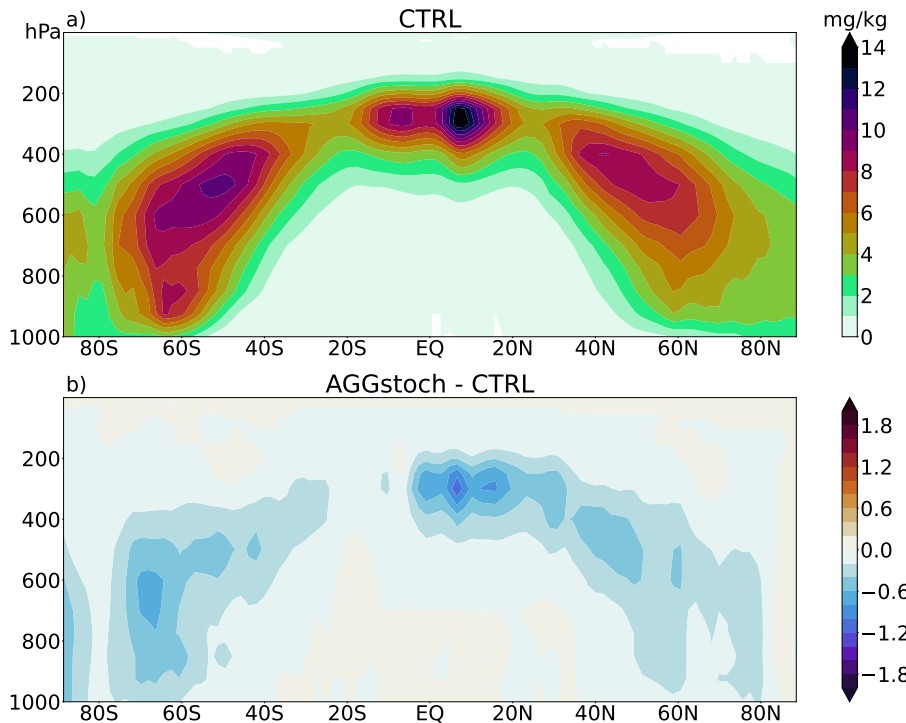

**Figure 6.** Zonally averaged multiyear mean of cloud ice (mg/kg). a) control run of the ICON-AES; b) Difference between the new stochastic aggregation parameterization and the control run (adapted from (Trömel et al., 2021)).

the good agreement in the distribution pattern, there are some regional differences. In 300 hPa, the maxima over the northern part of North America and central Africa are more intense in the DARDAR data than in the ICON data. At 400 hPa ICON underestimates the cloud ice variance compared to DARDAR. At lower altitudes the cloud ice variance is higher in the ICON data in most of the mid-latitudes compared to the DARDAR data. ICON underestimates the cloud ice variance in the tropics especially in lower regions, since ICON shows less cloud ice in the tropical region at the same pressure levels. However, we should keep in mind, that the measured variance contains uncertainties regarding the method of filtering precipitation and convection. Additionally the modeled cloud ice variance makes an assumption of a distribution, while the DARDAR data shows the the variance from discrete measurements. Despite these discrepancies, we conclude that the simple uniform distribution of cloud ice as it is written in Eq. (14) basically captures the measured distribution of cloud ice in each grid box. Therefore it is suitable for use in the aggregation parameterization.

As described in Section 2.2, with the help of this cloud ice distribution a new specific cloud ice mass is used for the aggregation parameterization. Figure 6 shows its influence on the zonally averaged cloud ice adapted from (Trömel et al., 2021), where the same tuning parameters were used. The control run of the ICON model (CTRL) gives a maximum of cloud ice at levels of lower pressure over the tropics. There are also two maxima over the mid-latitudes between 600 hPa and 400 hPa, with a more pronounced maximum in the southern hemisphere than in the northern hemisphere. An overall reduction of cloud ice is visible

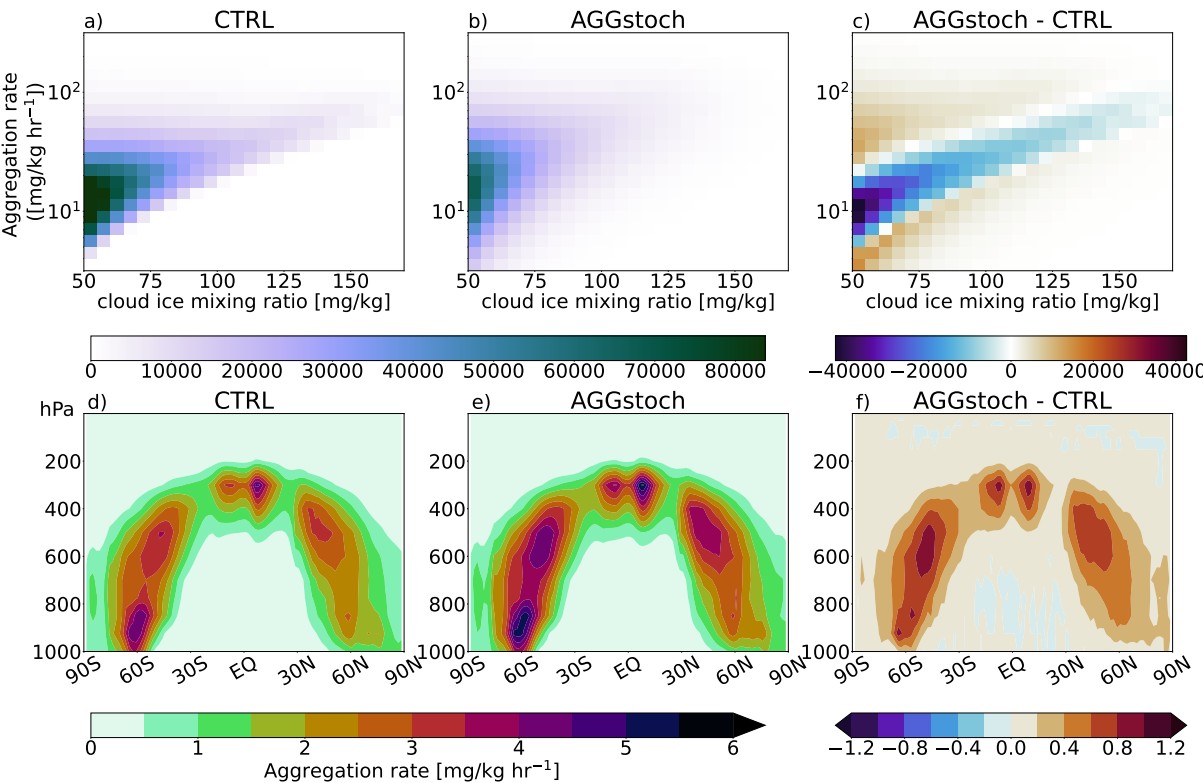

**Figure 7.** Joint histrograms of aggregation rate (mg/kg hr$^{-1}$) and cloud ice (mg/kg) for CTRL (a), AGGstoch (b) and the difference between each other (c); zonal averaged aggregation rate of multiyear-mean for CTRL (d), AGGstoch (e) and the difference (f)

due to the stochastic aggregation scheme (AGGstoch).

Since aggregation is a non-linear process, it becomes stronger due to taking a distribution of cloud ice into account. A stronger aggregation process leads to a higher conversion rate from cloud ice to snow. Therefore, more cloud ice is removed due to the process. Figure 7 shows the effect of using the stochastic approach instead of the default parameterization directly for the

220 aggregation rate in a joint histogram and the corresponding mulit-year zonal averaged aggregation rate. While the CTRL simulation produces an intense maximum of the aggregation rate around ca. $30\,\mathrm{mg\,kg^{-1}\,hr^{-1}}$, the AGGstoch run reveals a larger spread around a less intense maximum. The higher spread of the maximum stems from the randomly picked cloud ice mass for the aggregation process. Therefore, there are values which are much higher and much lower, respectively, than the original grid-box mean cloud ice mass. Instead of having this intense maximum of the aggregation rate, higher and lower aggregation

rates cases are visible. Therefore, the annual average over all of these aggregation rate cases yields a higher value for the AGGstoch run compared to CTRL run, caused by the non-linearity of the process, which is also visible in Figure 7f).

The microphysical processes in clouds are strongly interlinked with each other and react whenever one process rate changes. Figure 8 shows how the different cloud ice related process rates change using the stochastic aggregation parameterization. The

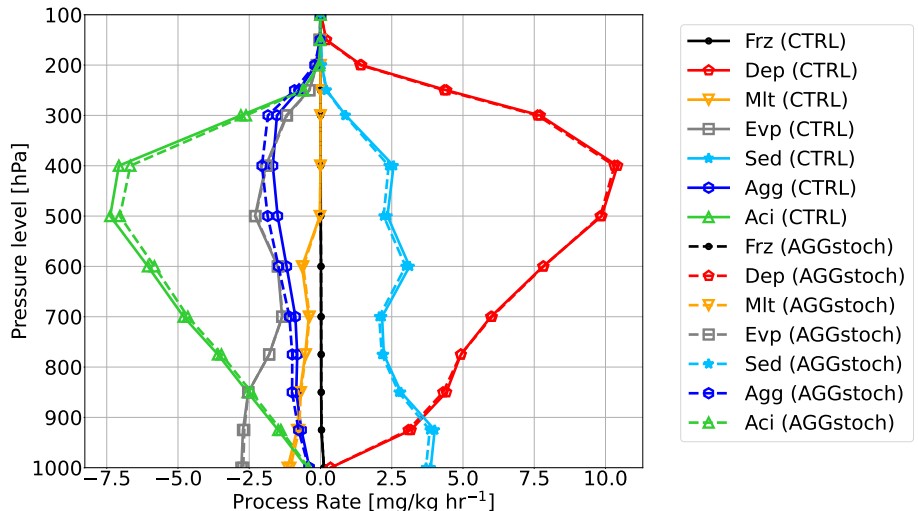

**Figure 8.** Vertical profiles of the global mean process rates which are related to cloud ice microphysical processes for CTRL (solid lines) and AGGstoch (dashed lines): (Aci), Freezing rate (Frz), Deposition rate (Dep), Melting rate (Mlt), Evaporation rate (Evp), Sedimenation rate (Sed), Aggregation rate (Agg) and Accretion rate. (transport terms are not included)

different vertical profiles of the global mean process rates are depicted. Negative process rates indicate a cloud ice loss, while positive process rates lead to a cloud ice gain. So, a more negative process rate is linked with a stronger cloud ice loss. It is visible, that the deposition rate is the most effective cloud ice gaining process. It is triggered by the Wegener-Bergeron-Findeisen (WBF) (Wegener, 1911; Bergeron, 1935; Findeisen et al., 2015) process. This process takes place if the temperature is between -35 $^o$C and 0 $^o$C and the $q_i$ exceeds a specific threshold (Giorgetta et al., 2013). The strongest negative process

rate is the accretion rate. The accretion rate (Aci) describes the growth of snow by collecting the surrounding ice crystals. Therefore, the accretion rate is strongly linked with aggregation rate, since the aggregation process produces the snow first. Aggregation and accretion describe the formation and growing of snow by coagulation of cloud ice and thus, they lead to a cloud ice reduction. The riming process (not shown here) is described like the linear accretion process and leads to increase in snowfall, but with the difference, that the equation depends on cloud water. However, the accretion rate is the more efficient

process compared to the aggregation. This plot confirms the higher cloud ice loss in the AGGstoch compared to the CTRL simulation due to the more negative aggregation rate. Moreover, due to the stronger aggregation rate at levels of high pressure, less cloud ice may be transported into lower regions. Hence, the sedimentation process slightly decreases. Due to the change in the aggregation process rate, accretion rate becomes less intense and leads to more cloud ice. The reduced accretion rate may be explained by the cloud ice loss due the stronger aggregation rate. Less cloud ice is collectable for growing snow flakes.

No significant changes are visible for the freezing, the deposition, the melting and the evaporation rate. Due to the stochastic method, the global mean aggregation rate is intensified by more than 20 % in the middle and upper troposphere. In contrast, the maximum change of global mean accretion rate is less than 8.9 %. Since the accretion rate is much higher, this smaller change leads to a compensation of the aggregation rate increase as described above. However, the increase of aggregation rate

by more than 20 % is significant. Due to the small change in the microphysical properties, no important change in multiyear global mean shortwave (+0.161 Wm$^{-2}$), longwave (-0.195 Wm$^{-2}$) and net-radiation (+0.03 Wm$^{-2}$) at the top-of-atmosphere is visible (not shown here). Overall, we found a cloud ice loss in the AGGstoch run of up to 5 %, but the reduction of cloud ice is compensated by a less intense accretion rate. The effect on radiation could be increased, if this stochastic approach was implemented in other processes, since there has to be a change of a factor 2 or more in microphysical process rates to see an effect on radiation (Michibata et al., 2020; Imura and Michibata, 2022). Using the new approach just for one single process makes it easier in the beginning to see the effect of changing one process, since all processes are connected. Besides from that, the aggregation process is the only non-linear cloud ice process rate in the ICON-AES. Since we focused on cloud ice related processes we just implemented the subgrid-scale approach in the aggregation parameterization. In future studies one can think about including additionally the subgrid-scale approach in the cloud water related processes, in order to see a stronger effect on radiation, cloudiness and precipitation as well.

As already mentioned, using a stochastic approach in non-linear process rates lowers the bias of the process rate. To create an unbiased aggregation rate (AGGsample), as described above, we make use of the entire cloud ice distribution function. Figure 9 shows a comparison of the averaged aggregation process rate at different $q_i$-bins and the 2D-histograms of CTRL, AGGstoch and AGGsample. AGGstoch shows a good agreement to AGGsample, while CTRL produces much lower process rates. Both AGGstoch and AGGsample produce a higher aggregation rate compared to CTRL, especially for higher $q_i$ values. Figure 9b) gives the corresponding joint histogram of CTRL and c)-d) the difference histogram to CTRL of cloud ice and the aggregation rate. AGGstoch shows a higher spread in the dirstribution, as already mentioned, while in AGGsample the distribution is shifted towards higher aggregation rates. However, both methods produce in average higher aggregation rates for different $q_i$ sections. The main difference between these methods is, that AGGsample needs an additionally integration over $q_i$ for the calculation of aggregation rate, while AGGstoch needs no extra computational time. Therefore, the described stochastic method, which is used in AGGstoch, allows an improvement of the process rate bias against CTRL.

## 4 Summary

We introduce a stochastic approach of the aggregation parameterization in the ICON-AES by considering, consistently with the models' cloud scheme, a uniform distribution function of cloud ice and randomly choose a cloud ice water content from this distribution function for each grid-box and time step. The chosen distribution function of cloud ice is evaluated with the help of a combined Lidar/Radar data set (DARDAR). An estimate of the precipitating and convective cloud ice mass is removed from the data set, in order to allow a more consistent comparison to the cloud ice in the model. The comparison of the simulated and the observed cloud ice variance shows a good agreement in pattern, with just a few regional differences.

Overall we show that the uniform distribution function of cloud ice is usable for the microphysical process rates e.g. the aggregation. From this uniform distribution function a randomly chosen cloud ice value is implemented into the non-linear aggregation rate in order to represent subgrid-scale variability. Due to this stochastic method, the aggregation rate is intensified on average, since aggregation is a non-linear process. As a result, more cloud ice is transformed to snow, which leads to a cloud

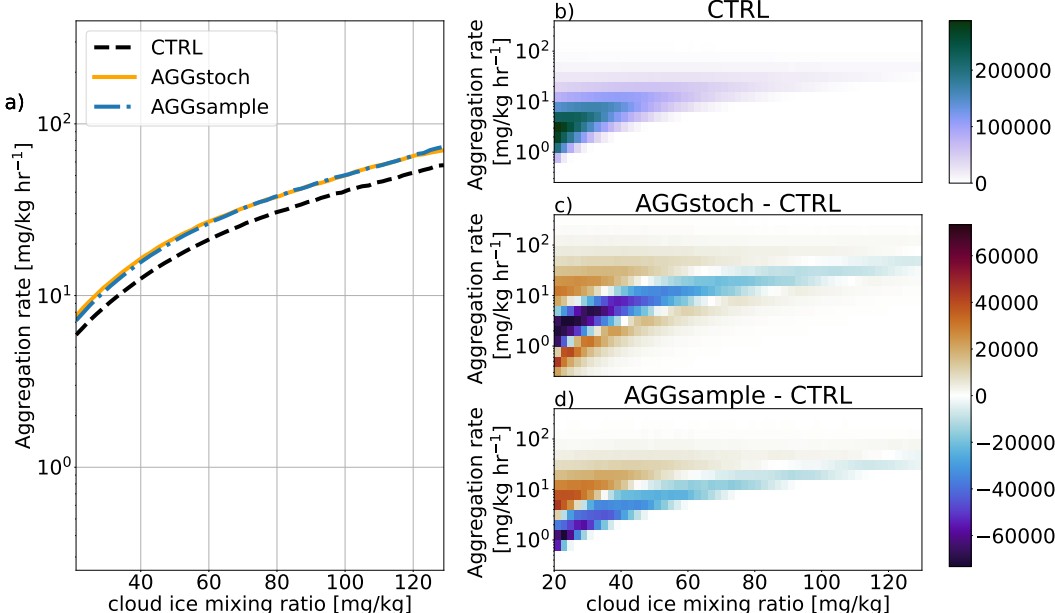

**Figure 9.** Averaged aggregation rate at different cloud ice mixing ratio bins and the corresponding 2d histograms for CTRL, AGGstoch and AGGsample

ice loss. However, the decrease in the accretion rate, that results from using the stochastic aggregation scheme, acts against the more intense aggregation rate. Therefore, the cloud ice loss is not as strong as expected. This indicates that changing only one of the two process rates of snow formation does not lead to a change as large as one might have initially expected because

aggregation and accretion interact very strongly with each other. However, the effect of taking subgrid-scale variability into account for process rates has a significant impact on the microphyscial process (20 % stronger global averaged process rate). An unbiased process rate is calculated integrating the aggregation rate over the entire cloud ice distribution function. The new stochastic method shows a better agreement of the aggregation rates with the unbiased method compared to the control run. It follows, that the new method lowers the bias of the aggreagtion rate, which does not need additional computational

time. Therefore, this study suggests that using a stochastic approach for microphysical process rates helps to improve the representation of clouds and precipitation processes in global climate models.

*Data availability.* The ICON-AES data are stored at the German climate computing center (DKRZ). The 2B-CLDCLASS CloudSat data and DARDAR data are downloaded from the ICARE Data and Services Center (https://www.icare.univ-lille.fr/asd-content/archive/)

*Code and data availability.* The source code of this study is available at https://zenodo.org/records/10788230. Updates of the code can be found at https://github.com/shoernig/code_subgrid_scale. Information about the licence and the respective README files are also included. Access to ICON source code can be obtained at (https://icon-model.org/).

## Appendix A: Variance of cloud ice

Here, the calculation of cloud ice variance, which is used in the ICON-AES, is given step by step. The general equation is written as follows:

$$\sigma_{q_i}^2 = \int_0^{2\Delta q_i} (q_i - \overline{q}_i)^2 \, \text{PDF}(q_i) \mathrm{d}q_i \tag{A1}$$

Separating the integral in the cloud-free part, which is expressed as a Dirac function, and the cloudy parts yields:

$$\sigma_{q_i}^2 = \int_0^{2\Delta q_i} \left[ \underbrace{(1-C)\,\delta(q_i=0)}_{\text{cloud-free part}} + \underbrace{C\,\frac{1}{2\Delta q_i}}_{\text{cloudy part}} \right] (q_i - \overline{q}_i)^2 \mathrm{d}q_i \tag{A2}$$

Solving the Dirac function of the cloud-free part yields:

$$\sigma_{q_i}^2 = (1-C)\,\overline{q}_i^2 + \frac{C}{2\Delta q_i} \int_0^{2\Delta q_i} (q_i - \overline{q}_i)^2 \mathrm{d}q_i. \tag{A3}$$

Solving the integral with $\overline{q}_i = C\Delta q_i$ (see Eq. (8)):

$$\sigma_{q_i}^2 = (1-C)\,\overline{q}_i^2 + \frac{C}{6\Delta q_i} \left[ (2\Delta q_i - \overline{q}_i)^3 + \overline{q}_i^3 \right]$$

$$\sigma_{q_i}^2 = (1-C)\,(C\Delta q_i)^2 + \frac{C}{6\Delta q_i} \left[ (2\Delta q_i - C\Delta q_i)^3 + (C\Delta q_i)^3 \right]$$

$$\sigma_{q_i}^2 = (1-C)\,(C\Delta q_i)^2 + \frac{C}{6\Delta q_i} \left[ 8\Delta q_i^3 - 12 C\Delta q_i^3 + 6C^2\Delta q_i^3 \right]$$

$$\sigma_{q_i}^2 = (\Delta q_i)^2 \cdot \left( \frac{4}{3}C - C^2 \right). \tag{A4}$$

Replacing $\Delta q_i$ with Eq. (10):

$$\sigma_{\overline{q}_i}^2 = (\overline{q}_i^c)^2 \cdot \left( \frac{4}{3}C - C^2 \right). \tag{A5}$$

In the end, the variance of cloud ice just depends on the cloud cover and grid-box mean of in-cloud ice. So, the variance, which is calculated for the aggregation process, can be checked directly for the specific amount of cloud ice, which is available for the aggregation.

*Author contributions.* SD and JQ conceived this study. JK gave expertise on how to run the model. OS gave advise about the satellite data. JK, JQ and MS helped by checking the theoretical approach of this study. SD prepared the article with contributions from all co-authors.

*Competing interests.* The authors declare that they have no conflict of interest.

*Acknowledgements.* We gratefully acknowledge the funding of the German Research Foundation (Deutsche Forschungsgemeinschaft, DFG) to in its priority program "Fusion of Radar Polarimetry and Atmospheric Modelling" (SPP-2115, PROM), specifically the project PARA (FKZ QU 311/21-1). JQ further acknowledges funding by the EU Horizon 2020 project "FORCES" (GA 821205). This work used resources of the German Climate Computing Centre (Deutsches Klimarechenzentrum, DKRZ) granted by its Scientific Steering Committee (WLA) under project ID. We thank the AERIS/ICARE Data and Services Center for providing access to the DARDAR and 2B-CLDCLASS data used

in this study. We thank Takuro Michibata and a anonymous reviewer for comments that led to improvements in the analysis and manuscript

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
