# Peer review of "Subgrid-scale variability of cloud ice in the ICON-AES 1.3.00"

_Geoscientific Model Development, 2022_

## Author Comment (AC1)

Dear Editor,

We thank the reviewer for thoroughly reviewing our work and providing us with the chance to improve our study. We have carefully considered and incorporated your comments and suggestions into the revised manuscript. We are confident that the alterations made, including those you highlighted, have effectively addressed the concerns.

Please find below the modifications made at the manuscript. The review comments are in black, our point-by-point responses are in red color, and the changes to the manuscript are in blue color.

Sincerely,

Sabine Doktorowski on behalf of all coauthors

Responses to referee comments on the Manuscript "Subgrid-scale variability of cloud ice in the ICON-AES 1.3.00", by Doktorowski et al.

**General Comments**

This article is well structured and clear, and the language is fluent and precise. While there is a significant body of literature detailing methods for and consequences of accounting for subgrid variability of cloud water content in atmospheric model radiation calculations and warm rain processes, to my knowledge there are no previous studies on accounting for subgrid variability of cloud ice in microphysical process calculations. The methods are clearly described, and the results support the interpretation and conclusions.

My main concern with the article is that I think this is quite a minor advance in modelling science – in the authors' own words, this produces "no important change in radiation". The title of the article suggests a broad analysis of subgrid scale variability of cloud ice in the model, but the study is limited to a single process. I think this paper would be more signicant if it considered other processes. Do any of the other ice microphysics processes in the model have a nonlinear dependence on ice water content? If so, can this method be extended to those processes? What about the model radiative transfer calculations? Do they account for subgrid-scale variability of cloud ice and if so, do they use the same variability as used for ice aggregation here? We thank the reviewer for this point. The aggregation is the only cloud ice related microphysical process, that is not linear. Therefore, we just implemented it in the aggregation process. Besides form that, we wanted to show the effect on one single process, to quantify, how big the effect on the process rate is. We agree, that we could increase the effect, if it would be possible to extend this method to other cloud ice processes. Since the model doesn't include any subgrid-scale approach for the radiation, this could be part of further studies to implement such an approach to the radiation scheme. Additionally, one can think of using this method for cloud water processes, which lead to a gain of cloud ice (e.g. freezing). To goal of this study is to show the effect of this stochastic approach, which doesn't need any further computational time, on the aggregation rate, which is a cloud ice related process. Since this method is so simple, it could be used for every non-linear process rate, also in other models, were maybe more cloud ice related processes are non-linear.

**Specific Comments**

1. It would be good to include analytically derived corrections (e.g., Morrison and Gettelman, 2008; Larson and Griffin, 2012; Boutle et al, 2014) in the discussion of how the effects of subgrid variability of cloud water content on microphysical process rates have been represented in previous studies. Although most of this literature focuses on warm rain microphysics rather than ice, I think it is relevant to the discussion. Is it possible to derive an analytical correction to

45     the ice aggregation rate? Thank you very much for this input. We included additional text in the introduction to highlight, that our focus is on cloud ice related processes in contrast to precious studies, which focused more on liquid water processes. While it is possible to do a numerical integration, here we focus on this stochastic approach, since it doesn't need additional computational time, as it is the case in e.g. Larson and Griffin (2012). Therefore, we included Figure 8 to show, how the comparison between the stochastic method and a sampled method, where we sampled over the entire

50     distribution, looks like. Since this plot shows a good agreement, we just want to go for this simple method.

Including the subgrid-scale effect in the autoconversion and accretion rate in warm clouds reduces the bias significantly and leads to an enhancement of the process rate (Boutle et al., 2014; Lebsock et al., 2013). Since previous studies mainly focus on warm rain formation processes (e.g., Morrison and Gettelman, 2008; Larson and Griffin, 2013; Boutle et al., 2014; Lebsock et al., 2013), it is also important to concentrate on snow formation effects, by taking subgrid-scale effects

55     into account.

2. Can you explain why you only apply a representation of the effects of subgrid variability of ice water to the aggregation calculation? Do other ice microphysical processes in ICON-AES depend nonlinearly on ice water content? Thank you for this question. Aggregation is the only non-linear process, which depends on $q_i$. And since we wanted to concentrate on cloud ice related processes, we just implemented the stochastic approach into the aggreagtion. The method can also

60     be extended to liquid water related processes. We added in the manuscript: Besides from that, the aggregation process is the only non-linear cloud ice process rate in the ICON-AES. Since we focused on cloud ice related processes we just implemented the subgrid-scale approach in the aggregation parameterization.

3. Does the ICON-AES radiation scheme include the effect of subgrid variability of ice water content on radiative fluxes and heating rates? If so, what does it use for the subgrid variability? Assuming the radiation scheme does not already

65     use ice cloud water content variability that is consistent with the cloud scheme, what difference would this make? The radiation scheme in the ICON-AES doesn't include subgrid variability of ice water content beyond the vertical overlap of cloudy layers. But it is indeed an interesting point. One can try to implement an subgrid-scale approach in the radiation scheme in further studies.

4. I am not convinced that the comparison between DARDAR and the model is particularly useful. I think it would be

70     more useful to compare aggregation rates calculated using the cloud scheme subgrid ice variance, aggregation rates calculated using the "true" variance (i.e., values derived from DARDAR) and aggregation rates calculated using only the mean value (i.e., a similar analysis to that done for figure 8). This could be an additional plot and, in my opinion, would better demonstrate the utility of the cloud scheme subgrid ice cloud variance. We thank you for this interesting point. Since aggregation rates are not retrieved from the satellite measurements, we decided to compare the cloud ice

75     variances, because it is a more straight forward comparison. Of course there are still some uncertainties and differences between modeled and the "true" variance. Nevertheless, using the cloud ice variance from DARDAR and putting this into the aggregation parametrization makes it more unclear in our opinion, since the aggregation rate depends on other conditions. Therefore, we think it is a better way instead of putting additional uncertain conditions in the comparison, we directly tried to compare the variances. However, we do consider the differences between the assumed cloud ice

80     distribution and the real cloud ice distribution. We added to the main text: However, we should keep in mind, that the measured variance contains uncertainties regarding the method of filtering precipitation and convection. Additionally the modeled cloud ice variance makes an assumption of a distribution, while the DARDAR data shows the the variance from discrete measurements. Despite these discrepancies [...]

5. I'm not sure how fair it is to compare the variance along a 1D line through a cloud (i.e., what DARDAR sees) with that

85     in a 3D gridbox (ICON-AES). For example, Hill et al (2015) estimated that the standard deviation of water content in a 2D cloud would be approximately 1.3 times larger than that in a 1D cross-section through the cloud. Can you comment on how this might affect your DARDAR – model comparison? Thank you for recommending this study from Hill et al (2015). It is indeed a very important point to highlight, that there are uncertainties comparing a 1D track with a 3D gridbox. We included a brief discussion in the method to include your suggested study.

90     Additionally, the initial satellite data are measured on a 1D curtain, while the model uses 3D grid boxes. Hill et al.

(2015) calculated a measure of the difference in standard deviation considering a 2D cloud field compared to a 1D cross-section. They estimated a 30% larger standard deviation in 2D fields compared to the 1D track. Therefore, we should also consider, that this has an effect also on our the cloud ice variance calculation. However, there are limitations in availability of satellite data. Therefore we tried to be consistent as possible in the comparison between simulations and observations.

6. I'm not entirely convinced that the way that the DARDAR data is sampled (I.e., removing convective and precipitating ice) achieves the aim of making it more consistent with the model. It would be interesting to see how much difference removing convective and precipitating ice makes to the variance of ice water content calculated from DARDAR. It would also be interesting to try some alternative comparisons between the two. For example, precipitating ice is removed from DARDAR based on a surface precipitation flag. What difference would it make if you removed ICON points with nonzero surface precipitation from the comparison? We thank you for this interesting suggestion. We agree, that alternative comparison between the model and the satellite data would be very interesting. We decided to use this method, since we wanted to compare the cloud ice variance and the cloud ice, which is directly used for the aggregation parameterization. The cloud ice, which is used for microphysical processes, doesn't contain any snow particles. In order to have a proper comparison between satellite cloud ice and the cloud ice used in the aggregation, we had to adjust the satellite data in the way, that we tried to remove precipitation and convection based on the study of Li et. al. (2012). Further studies can be focused more on what is the best way to compare modeled and observed cloud ice.
In order to compare modeled cloud ice with observations, alternative methods are possible (e.g. removing ICON non-zero surface precipitation points,...). Since we want to evaluate the cloud ice distribution, that is used for the aggregation, we had to adjust the DARDAR data in order to find the most consistent way.

7. For Figure. 2 and 3 a third column showing the difference between the two would be really useful. If necessary, you could plot this at lower resolution to reduce noise. Thank you very much for this advise to create a third column with differences to Figure 3 and 4 (now Figure 5). Since you wrote Figure 2 and 3, we also added the difference to Figure 2.

[Figure]

**Figure 2.** Zonally averaged annual mean of ice water content (mg/kg) from the DARDAR data set. a) total ice mixing ratio ($q_{i,total}$), which includes cloud ice from any clouds, and precipitating ice; b) cloud ice mixing ratio ($q_i$), where precipitating and convective cloud ice are removed; c) difference between $q_{i,total}$ and $q_i$.

8. You state that there is "No important change in radiation". Is this also true for precipitation rates? If this doesn't lead to any important changes then is it worth implementing in the model? Interesting point. The implementation of the subgrid-scale approach is a more accurate way to describe the cloud. Since it needs no further computational time, it is an easy way to bring a more realistic description of cloud ice to the model. We mainly wanted to show, what the effect of this statistic approach is. Since this method can be extended to other processes (also cloud water processes), we first needed a study on how the effect is for one single process rate, even there is no strong signal in radiation and precipitation.

9. I think it may better to have figure 8 and the discussion of the change in aggregation rate before figure 5, to demonstrate that the stochastic method does a good job or reproducing the unbiased aggregation rate before showing the effect of the stochastic method in the model. Thank you for this useful advise. Instead of rearranging the result part, we hint at this part already in the method and introduction part, to focus on it.
Introduction part: As an additional evaluation, we compare an unbiased process rate with the stochastic approach in order to investigate, how well this simple method performs.
Method part: To compare the current, biased aggregation rate ($Q_{agg}(\overline{q}_i)$) with the unbiased process rate [...] with the integral over the entire distribution of $q_i$. This comparison will be shown in the last part of the results.

[Figure]

**Figure 3.** Multiyear mean of the cloud ice mixing ratio (kg kg$^{-1}$) at four different pressure levels calculated for the DARDAR data (a,d,g,j), the ICON data (b,e,h,k) and diference between ICON and DARDAR (c,f,i,j).

10. The paper is quite concise already but could be made more so by removing table 1, which in my opinion does not add a great deal. Thank you for this suggestion. We removed the table from the main text. We just kept the text with the percentage numbers without showing the table, since we think it is useful to quantify the change in aggregation and accretion rate.

**Technical Corrections**

1. L5: "For a realistic comparison . . . removed from the observational data set." I think this is more technical detail than is needed for an abstract and suggest removing this sentence. We removed the sentence from the abstract.

2. L6: "The global patterns of . . . despite some regional differences". This is a bit vague, and I would add some more specific details e.g., quantify the % difference between the model and observations. Good point. We extended the sentence as follows:The global patterns of simulated and observed cloud ice mixing ratio variance are in a good agreement, despite an underestimation in the tropical regions, especially at lower altitudes, and an overestimation in higher latitudes from the modeled variance.

3. L39: "Instead of taking a grid-box mean . . . with a randomly chosen cloud ice mass". I think it might be clearer to rewrite this as: ". . . with a cloud ice mass randomly chosen from the distribution of cloud ice mass assumed in the

[Figure]

**Figure 4.** Same as figure 3 but for cloud ice variance (kg$^2$ kg$^{-2}$)

cloud scheme". We changed the sentence as follows: Instead of taking a grid-box mean in-cloud ice mixing ratio for the non-linear aggregation parameterization, we feed the process rate with a cloud ice mass randomly chosen from the distribution of cloud ice mass assumed in the cloud scheme.

4. L57: Change "with an instantaneously output" to "with instantaneous diagnostics output" We changed this phrase.

5. L86: Change "which allows biases" to "which introduces biases". We changed it towards your suggestion.

6. L171: "see the supplement material" should be "see the supplementary material". However, I was not able to find any supplementary material to view anyway. Thank you for finding this mistake. We removed the supplementary material after the editors comments. We now removed this phrase from the main text.

7. L177: typo "at the same lebvels".
We corrected the typo.

8. L179: Change "Therefore it is usable for" to "Therefore it is suitable for use in". We changed the sentence towards your suggestion

9. L188: typo: "averaged aggregattion rate" We corrected the typo.

---

## Author Comment (AC2)

Dear Editor,

We thank the reviewer for thoroughly reviewing our work and providing us with the chance to improve our study. We have carefully considered and incorporated your comments and suggestions into the revised manuscript. We are confident that the alterations made, including those you highlighted, have effectively addressed the concerns.

Please find below the modifications made at the manuscript. The review comments are in black, our point-by-point responses are in red color, and the changes to the manuscript are in blue color.

Sincerely,

Sabine Doktorowski on behalf of all coauthors

Responses to referee comments on the Manuscript "Subgrid-scale variability of cloud ice in the ICON-AES 1.3.00", by Doktorowski et al.

General comments:

This study developed a new scheme for the cloud ice aggregation process, which considers subgrid-scale variability based on a stochastic approach. The authors introduced the scheme into ICON-AES, and evaluated the impact on cloud ice representation against the CloudSat/CALIPSO-based DARDAR product. The new scheme enhances the aggregation rate to reduce cloud ice, which is compensated by decreasing the accretion of ice. Although it was reported that the impact of the new scheme on cloud and radiation fields is not significant, the new approach could provide an important advance in the subgrid-scale representation of ice clouds in future studies.

I feel that the appropriately revised manuscript after the authors include the minor suggestions below may be suitable for publication in Geoscientific Model Development.

Specific comments:

Introduction: There are some previous studies for correcting microphysical process rates by considering subgrid-scale variabilities of hydrometeors, though the approach and target differ from the present study. For example, the analytical formulas for modifying autoconversion and accretion rates in the liquid phase have been developed (Lebsock et al., 2013; Boutle et al., 2014), but I do not know any methods for ice-phase clouds. Adding brief discussions for these previous studies would be helpful for readers and for clearing up the novelty.

Thank you very much for this point, which highlights more the importance and novelty of this study.

Including the subgrid-scale effect in the autoconversion and accretion rate in warm clouds reduces the bias significantly and leads to an enhancement of the process rate (Boutle et al., 2014; Lebsock et al., 2013). Since previous studies mainly focus on warm rain formation processes (e.g., Morrison and Gettelman, 2008; Larson and Griffin, 2013; Boutle et al., 2014; Lebsock et al., 2013), it is also important to concentrate on snow formation effects, by taking subgrid-scale effects into account.

Line 57-58: Could you describe model configurations and parameterizations in more detail? For example, "... for a period from 2005 to 2009 ..." may imply nudged configuration, right? Please also add information on the spin-up period and model time-step.

Thank you for the suggestion. We added the informations to the method part: All runs were performed for six years with prescribed sea surface temperature and sea ice boundary conditions for a period from 2004 to 2009 without with instantaneous output every six hours by using a model time step of 10 minutes, a horizontal resolution of 160 km and 47 vertical hybrid sigma levels up to 80 km height (Crueger et al., 2018; Giorgetta et al., 2018). To avoid any effect of the model spin-

50    up, we ignored the first year from the model results. Therefore all multiyear averages were done for the time period 2005-2009.

Line 57: "with an instantaneously output every ..." => "with instantaneous output every ..."
We changed the phrase to your suggestion.

55   Line 60-61: Could you add more specific information about the diagnostic cloud cover scheme?
Thank you for the suggestion. We added to the main text:
The final equation for C, which is used in the model, is given by

$$C = 1 - \sqrt{1 - \frac{r - r_0}{r_{sat} - r_0}}, \qquad\qquad (1)$$

where $r$ is the relative humidity, $r_{sat}$ is the saturation value (= 1) and $r_0$ is a function of pressure and depends on two different
60   tuning parameters ($r_{top} = 0.8$ and $r_{surf} = 0.968$), which defines the condensation threshold.
Line 62: How does the model treat precipitating hydrometeors? As far as I know, ECHAM6 physics includes prognostic rain and snow based on Sant et al. (2015), or optional? Please add a brief description of the treatment of precipitation (diagnostic or prognostic) because ice microphysical processes, which is the main focus of this paper, depend strongly on how models treat snowflakes (Michibata et al., 2020). Thank you for this comment. It is indeed very important to add informations on how
65   the model treats snowfall. Therefore we added to the method part: "The ECHAM6 physics includes diagnostic rain and snow profiles in the columns. It is not transported by advection"

Equations (4) and (5): Typo? Please check the symbol "X" in equation (4), and symbols "p0" and "p" in equation (5).
Thank you for checking the equation so carefully. We corrected the symbols in equation (4) ($\rho_0$ and $\rho$) and we added "X" in
70   equation (5).
Line 132-134: Consider adding references for CloudSat and CALIPSO missions (e.g., Stephens et al., 2002; Winker et al., 2010).
Winker, D. M. et. al.: The CALIPSO Mission, B. Am. Meteorol. Soc., 91, 1211–1230, https://doi.org/10.1175/2010BAMS3009.1, 2010.
75   Stephens, G. L. et al.: THE CLOUDSAT MISSION AND THE A-TRAIN, B. Am. Meteorol. Soc., 83, 1771–1790, https://doi.org/10.1175/BAMS-83-12-1771, 2002.
Thank you for the literature suggestion. We added both references to the main text: To evaluate the results, a combined global ice water product of the CloudSat (Stephens et al., 2002) and Cloud-Aerosol Lidar and Infrared Pathfinder Satellite Observations (CALIPSO) (Winker et al., 2010), (...)
80

Line 141-145: I do not have a complaint about the analysis method, but I believe that the comparison between model and observation without separating clouds and precipitating ice is more straightforward. Ice-to-snow conversion is continuous and therefore CALIPSO retrievals cannot separate between cloud ice and snowflake even though using the "precipitation flag". Noting possible uncertainty would be valuable for readers. And again, how does the model treat snowfall?
85   Thank you for raising this. It's indeed an interesting point. The idea of separating cloud ice and snowfall is, that we can compare the cloud ice variance from the model directly with the satellite observations. The cloud ice variance from the model is directly calculated before aggregation started and from the cloud ice, which is available for the aggreagtion. Since the model separates cloud ice and snowfall, we decided it is necessary to use the same method for the satellite data. We agree, that using the precipiation flag in order to remove snowfall from the data set is not completely perfect and therefore, there are still some
90   uncertainties comparing model data and satellite data. We added more text to highlight, that there are still some uncertainties with this methods. Additionally we added, how the model treats snowfall in the model to the method part.

Line 150: "... and cloud ice mixing ratio (qi)." => "... and cloud ice mixing ratio (qi) obtained from the DARDAR product."
We added "obtained from the DARDAR product" to the main text.
95

Line 151: "higher levels" and "lower levels": This is unclear. Please clearly state by adding pressure levels. Thank you very much for this advise. It was indeed unclear and we changed every "high/low level" cases to a more clear description.

Line 160: The result section starts with "2.4". Please correct the editorial error in the LaTeX commands. We corrected the error in the LaTex document.

Line 168-169 and Figure 3: Although the model seems to perform well in representing cloud ice distribution, how about column integrated water path (CIWP)? I am slightly concerned about how much the model overestimates CIWP, which directly affects process rates.
Thank you for this point. We included an additional CIWP plot to show the how the model performs in comparison with DAR-DAR.

[Figure]

**Figure 4.** Multiyear cloud ice water path (CIWP) for the DARDAR data and ICON data and the difference between DARDAR and ICON

The cloud ice water path (CIWP), which is the column integrated ice water content, is given in figure 4. ICON overestimates the CIWP in the middle and higher latidues, while it underestimates the CIWP in the tropics. As it was already visible in the figure 3 DARDAR shows larger cloud ice values down the lower altitudes over the tropics, which leads to an increased CIWP compared to the modeled CIWP. Especially in the midlatidues at higher pressure fields the model tend to overestimate the cloud ice. One should keep in mind, that the way the DARDAR data is filtered to get the CIWP or $q_i$, which is comparable with the model data, is not perfect.

Line 169: Start a new paragraph from "Figure 4 shows . . .". We started a new paragraph.

Line 177: Please correct the typo "lebvels". We corrected the typo.

Line 184: In the caption of Figure 5, the CTRL result was from Tromel et al. (2021). Just to check, were the tuning parameters between CTRL and AGGstoch the same? Please consider adding a note to the main text or figure caption. Thank you for raising this up. We checked all of the tuning parameters and they are the same. We added a note in the main text. "Figure 5 shows its influence on the zonally averaged cloud ice adapted from (Trömel et al., 2021), where the same tuning parameters were use"

Line 201-203: How does the model represent the riming process and Bergeron-Findeisen process? These microphysical processes are also the important source terms for cloud ice and thus snowfall (Gettelman, 2015; Michibata et al., 2020).

Line 206: Insert comma, between "aggregation process rate" and "accretion rate".
We added a comma to this sentence.

Line 212: Typo "8,9%" => "8.9%" We changed the punctuation.

Line 214 "no important change in radiation is visible (not shown here)": Consider adding changes in global mean shortwave, longwave, and net radiation at the top-of-the-atmosphere, which would be helpful as a more intuitive metric. Thank you very much for this comment. We included your suggestions with the global mean values in brackets: Due to the small change in the microphysical properties, no important change in global mean shortwave (+0.161 Wm-2), longwave (-0.195 Wm-2) and net-radiation (+0.03 Wm-2) at the top-of-atmosphere is visible (not shown here)

Line 233-234 "Overall we show . . .": Start a new paragraph here. We started a new paragraph.

Line 241-242: I feel that a 20% change in the process rate is not significant enough to largely affect radiation and cloud fields. Some studies often show that significant changes in cloud, precipitation, and radiation require modifications of microphysical process rates by a factor of 2 or more (e.g., Imura and Michibata, 2022). In the present study, the authors applied the stochastic method to the aggregation process alone, but it could be extended to other ice microphysics as well (e.g., riming process, accretion among droplets, crystals, and snowflakes) in future studies. When such a framework is incorporated into the model, the impact of the scheme on radiation would be larger than the current model. Please consider adding such brief discussions to the revised manuscript.
Thank you very much for this input. It is indeed a very important point. We added a brief discussion: The effect on radiation was increased, if this stochastic approach could be implemented in other processes, since there has to be a change of a factor 2 or more in microphysical process rates to see an effect on radiation (Michibata et al., 2020; Imura and Michibata, 2022). Using the new approach just for one single process makes it easier in the beginning to see the effect of changing one process, since all processes are connected. Beside from that, the aggregation process is the only non-linear cloud ice process rate in the ICON-AES. Since we focused on cloud ice related processes we just implemented the subgrid-scale approach in the aggregation parameterization. In future studies one can think about including additionally the subgrid-scale approach in the cloud water related processes, in order to see a stronger effect on radiation, cloudiness and precipitation as well.

---

## Author Response (AR2)

Dear Editor,

First of all, we thank the Editor for their careful reading and the opportunity to improve our study.

Editors points:

1. In the "data availability" section, please remove the "available upon request to the corresponding author" remark. We removed the phrase.

2. The ICON licensing information page referenced in the "Code and data availability" section seems obsolete (it starts with a "This site is deprecated!" header) - please update the links and license statement accordingly. We replaced the old link with https://icon-model.org/

3. In equations: We implemented the suggested changes.

    – avoid using italics for text subscripts (e.g., "total", "sat", "ice", "i") - some of these are given with italics, some without;

    – equation formatting for (A2) could likely benefit from some adjustment of the bracket size (this could help: https://tex.stackexchange.com/questions/23161/using-underbrace-without-having-left-and-right-scale)

4. Please provide Fig. 1 in vector graphics format We uploaded the Figure as a vector graphic

5. In references: We cahnged the references.

    – fix URL (repeated https://doi.org/ prefix) in: Boutle et al. 2014, Hill et al. 2015, Larson et al. 2013, Lebsock et al. 2013 & Winker et al. 2010;

    – correct bibliographic data for Wegener 1911 - currently it leads to a 1912 review of this book, perhaps this URL could be better: http://worldcat.org/oclc/39667532

    – if providing the worldcat url for Wegener's book, Bergeron's book can be pointed to with https://worldcat.org/oclc/31921934

    – for Findeisen's work, the following DOI can be added: http://doi.org/10.1127/metz/2015/0675

6. Please release an updated cleaned-up code archive with the following files excluded: .git folder and its contents, all __pycache__ folders and their contents, files with tilde-sufixed names (these are also part of the referenced repository, and can safely be removed). For information on how to automate git repo archival on Zenodo, see: https://docs.github.com/en/repositories/archiving-a-github-repository/referencing-and-citing-content We added a new Zenodo link with the a cleaned-up version: https://zenodo.org/records/10788230

7. Other:We changed the manuscript towards your suggestion

    – page 15, line 289: change "doesn't" into "does not"

    – page 15, line 293: correct "dta" typo

We additionally included the following sentence to the acknowledgments:

We thank Takuro Michibata and a anonymous reviewer for comments that led to improvements in the analysis and manuscript.